# Traumatic Optic Neuropathy: Update on Management

**Mohammad Reza Hosseini Siyanaki [1],\***, **Mohammed A. Azab [2]** and **Brandon Lucke-Wold [1]**

1   Department of Neurosurgery, University of Florida, Gainesville, FL 32608, USA
2   Department of Neurosurgery, University of Cairo University, Cairo 12613, Egypt
*   Correspondence: mohammadhosseini@ufl.edu; Tel.: +1-(352)-273-6990

**Definition:** Traumatic optic neuropathy is one of the causes of visual loss caused by blunt or penetrating head trauma and is classified as both direct and indirect. Clinical history and examination findings usually allow for the diagnosis of traumatic optic neuropathy. There is still controversy surrounding the management of traumatic optic neuropathy; some physicians advocate observation alone, while others recommend steroid therapy, surgery, or both. In this entry, we tried to highlight traumatic optic neuropathy's main pathophysiologic mechanisms with the most available updated treatment. Recent research suggests future therapies that may be helpful in traumatic optic neuropathy cases.

**Keywords:** optic nerve; traumatic optic neuropathy; corticosteroids therapy; surgical management

## 1. Introduction or History

Traumatic optic neuropathy (TON) is a grave complication of closed head injury. It occurs in 0.7–2.5% of the population and is classified into direct and indirect TON [1].

Direct TON often results in complete vision loss and less chance of recovery than indirect TON. Direct TON usually occurs after bone fragments lacerate the optic nerve or when a concussion or contusion disrupts the optic nerve [2,3]. However, indirect TON usually develops due to blunt head or ocular trauma sustained via the oculofacial soft tissues and skeletal system to the optic nerve [4]. The visual loss may manifest several months after trauma; therefore, the diagnosis is always delayed [5]. TON management remains controversial; some physicians advocate observation alone, while others recommend high-dose corticosteroids, surgical optic canal decompression, or a combination. In this review entry, we try to highlight the main updates involving the management of TON.

## 2. Pathophysiology

Trauma can damage various parts of the optic nerve, including the intraorbital, intracanalicular, or intracranial parts [5]. TON can be caused by direct trauma such as penetration of a foreign body, displaced fracture fragment, or indirect trauma such as damage from transmitted concussive blows to the head. The orbital apex and optic canal transmit blunt trauma to the optic nerve [6]. The intracanalicular optic nerve is the most vulnerable to traumatic injury, but the intraorbital part is also at risk of injury [7]. The optic nerve's intracranial portion lies close to the falciform fold and is the next most common site vulnerable to injury [8]. Sphenoid fractures are usually present in half of the cases of TON, indicating that TON requires a high degree of force to occur [9]. Another injury is caused by open-air high-speed vehicles in which riders are subject to being thrown away from the vehicle or encountering a severe blow to the face, causing optic nerve avulsion [10]. Primary optic nerve injury is usually followed by retinal ganglion cell (RGC) axon shearing, where more than 80% die within one month. In contrast, secondary injury occurs due to swelling of the optic nerve due to mechanical trauma or vascular insult [11]. It is reported that ganglion cell axons are the site of injury in optic neuropathies, and based on that, animal models for TON were developed [12]. RGC injury is complex, and different treatments aim to protect these cells and help their regeneration. Investigating different ways to protect

the optic nerve is important as it cannot regenerate well in adults [13]. Moreover, different barriers, such as the blood–brain, blood–ocular, and aqueous barriers, make drug delivery to the optic nerve difficult.

### 2.1. Reactive Oxygen Species

RGCs undergo apoptosis in response to TON, and the possible mechanism is related to the production of inflammatory mediators and reactive oxygen species (ROS) [14]. Retinal degeneration is related to oxidative stress, and the endoplasmic reticulum and the mitochondria may play a role in that [15]. Ultrasound-induced optic nerve injury increased the concentration of ROS in the retina and caused optic nerve degeneration [16]. The systemic administration of G-CSF reduced the inflammatory response in optic nerve crush mouse models, reducing ROS and halting apoptosis among RGC [17]. In the murine traumatic brain injury model, RGCs and optic nerve degeneration were associated with oxidative stress [18]. CHOP, a downstream endoplasmic reticulum stress regulator of apoptosis, was upregulated in TON models within seven days after the insult [19]. ERO-1L, the protein active in both the oxidative stress and ER stress pathways, was elevated in experimental models of TON, indicating the prominent role of oxidative stress in pathogenesis [20]. A study by Saif Ahmed et al. reported that adenosine also functions as an anti-inflammatory molecule in retinal microglial cells through adenosine receptors [14].

### 2.2. Neuroinflammation

When an optic nerve is injured, microglia, astrocytes, and Müller cells are activated in the retina. Microglia become bipolar/rod-shaped and accumulate close to the deteriorating RGC axons in the nerve fiber layer (NFL). These microglia proactively internalized axons that were degenerating after ONI. The activated microglia take on an amoeboid appearance, engulfing the dead RGCs in the ganglion cell layer (GCL). Furthermore, the activated retinal microglia release pro-inflammatory cytokines (such as IL-6, and TNF-a), chemokines (CCL3, and CCL5), and ROS, resulting in urotoxic A1 astrocytes and increased neurotoxic factor production. Some pro-inflammatory cytokines are also produced by activated Müller cells in an injured retina. Eventually, a cumulative decline in survivors will gradually lead to permanent vision loss after ONI. [21,22] Macrophages also migrate to the injured area of the optic nerve after ONI. [23] Activated microglia and infiltrating macrophages have been demonstrated to degrade and remove myelin debris from the injured optic nerve, which is called Wallerian degeneration [24]. The pro-inflammatory cytokine TNF is an important modulator of immune cell function, as well as is correlated to neurodegenerative diseases [25,26]. Studies have shown that TNFR1 knockout mice are significantly protected from RGC toxicity during ONI [27].

### 2.3. The Role of Glutamate

Glutamate is an excitatory neurotransmitter whose levels are physiologically regulated in the retina by glutamate transporters. RGCs, photoreceptors, and bipolar cells release glutamate under physiological conditions [28]. N-methyl-D-aspartate (NMDA) receptors mediate the excitotoxic effects of glutamate on the RGCs [29]. The intraocular levels of glutamate rapidly increased after experimentally inducing retinal damage [30]. Following optic nerve crush injury, the glutamate transporter GLT-1 (EAAT-2) level was upregulated; there was a 3.9-fold increase in levels of GLT-1 mRNA compared with control eyes [31].

Edema associated with different optic neuropathies, including TON, is partly caused by the increased expression of aquaporins, especially Aquaporin 4 (AQP4) expression, on day one after optic nerve crush injury. Exposure of the cultured optic nerve astrocytes to glutamate significantly increased the expression of AQP4 [32]. The role of AQP4 in glutamate uptake by astrocytes was confirmed in different studies [33,34].

## 3. Optic Nerve Injury Classification

It is essential to distinguish between direct and indirect optic nerve injuries. Direct optic nerve injuries can be identified by eye exam or imaging techniques: transection of the optic nerve or its sheath, as well as orbital hemorrhage with emphysema. A more common type of traumatic optic neuropathy is indirect traumatic optic neuropathy, occurring when there is injury due to non-penetrating effects of trauma, such as hemorrhage, edema, or concussion. A head injury can damage the optic nerve due to impact force [7]. Approximately 0.5 to 5% of closed head injuries, 2.5% of midfacial fractures, and 10% of craniofacial fractures result in optic nerve damage [35].

In general, the blow causes unconsciousness, but the trauma is usually often mild, and the patient only seems disoriented. The visual field (VF) may be affected by a variety of defects. It takes 4–6 weeks for optic atrophy to develop.

### 3.1. Direct Injury

Direct optic nerve injury is often accompanied by severe orbital trauma, resulting in severe and immediate vision loss. On ophthalmoscopy, a hemorrhage ring can be seen around the optic disc. Imaging could verify the diagnosis. Ultrasound may be useful. Optic nerve transection is possible after midface or orbital trauma. An optic nerve-crossing bone fragment may be visible on a CT scan. There is no effective treatment. A diffuse or localized orbital hemorrhage can occur. This condition is associated with ophthalmoplegia and proptosis [36].

### 3.2. Indirect Injury

As a result of closed head injury, Crompton et al. observed hemorrhages in the optic nerve sheath (83%), nerve interstitium (36%), and ischemic necrosis in intracanalicular and intracranial nerve segments of 174 patients who died [8].

### 3.3. Injury Location

The anterior and posterior nerves might suffer direct and indirect injuries. Anterior TON, either direct or indirect, causes swelling of the optical disc with other signs of optic neuropathy. As a result, it may lead to retinal ischemia, anterior ischemic optic neuropathy, central retinal vein occlusion, choroidal rupture, or a composite of this situation [37].

There is no definitive prognosis for anterior TON caused by indirect insults. Blunt forces are typically responsible for indirect anterior TON. In addition to sudden shifts in intraocular pressure, rotational and translational forces have been linked to the pathogenesis of the damage. When this happens, the main pressure on the optic nerve is found opposite the impact site where the nerve meets the sclera [3]. Once the optic nerve has been damaged, there is no chance of benefiting from treatment. Unless there is a hemorrhage around the anterior optic nerve in the subdural or subarachnoid space, clinical findings can suggest such a condition and are verified with ultrasonography, computed tomography (CT) scanning, or magnetic resonance (MR) imaging [38]. Direct or penetrating trauma causing posterior TON often has a poor prognosis. The canalicular portion of the optic nerve is usually damaged by a posterior TON caused by indirect trauma. Depending on the angle of the force, the lines can cross both sides of the optic canal [39]. This can be attributed to several factors. First, blunt trauma to the frontal region has been shown to cause forces to travel posteriorly toward and sometimes into the canal, possibly due to the cone-shaped orbit that funnels energy to this area [40].

As a result of the initial injury, the optic nerve's canalicular portion is also susceptible to secondary injuries. The restricted surroundings of the canal expose this portion of the nerve as especially prone to swelling and hemorrhage, leading to ischemia, compartment syndrome, and secondary injury. Furthermore, the area is a drainage area, unlike other portions with rich anastomoses that receive blood, resulting in greater damage [41].

## 4. Diagnosis

### 4.1. Clinical Manifestation

Most acute or delayed visual function disorders associated with TON cannot be explained by anything other than a blunt craniofacial injury.

The following clinical features characterize TON:

(1) An immediate or delayed loss of vision after an injury;
(2) VF defects;
(3) Impaired color vision;
(4) Relative afferent pupillary defect (RAPD): occurs when the direct light reflex delays or disappears compared to the indirect light reflex; known as Marcus Gunn pupil [5,38]. Afferent disorders of the visual pathway are often associated with frontonasoethmoidal fractures [42].

Traumatic optic neuropathy is often associated with frontal (72%) or frontotemporal (12%) craniocerebral injuries. The most common causes are traffic accidents and falls [43].

An abnormal direct light reflex provides the most reliable diagnosis in patients with impaired consciousness. Most patients undergo a fundus examination, initially appearing normal if the lesion is not involving the retina and/or anterior optic nerve. However, the patient's optic disc will later become pale and atrophied [44].

The diagnosis of TON may be delayed by several combined craniofacial or other systemic injuries. The diagnosis may also be complicated by non-ON damage (damage to the eye or intracerebral visual circuits, etc.). In this case, the diagnosis of TON requires a comprehensive clinical assessment (ranging from ophthalmic to radiographic assessments) [45].

The majority of patients with TON present to the clinic with vision loss. The patient should undergo a complete ophthalmologic evaluation, including VF, VA, and fundus photography [46]. If the clinical ophthalmologic evaluation of the eye is not possible, pattern-reversal visual evoked potentials (P-VEPs) can be used. The P-VEP is useful in assessing visual function in children and infants as well as detecting nonorganic vision loss, nonorganic vision loss in complex patients, and optic neuropathy [47].

### 4.2. Imaging

The aim of imaging is to determine the extent of the damage to the visual pathway. The value of traditional radiographs is declining. The imaging modalities of computed tomography (CT) and magnetic resonance imaging (MRI) are preferred by some physicians when treating patients who are progressively visually impaired or who require therapeutic intervention [48,49]. Radiological investigations should be performed immediately for patients with head or oculofacial trauma and optic nerve damage (one or both decreased VAs, VF defects, and afferent pupillary defects) [50].

### 4.3. CT Scan

HRCT is a standard imaging technique for evaluating suspicious patients of TON because that can display a fracture of the ethmoid, sphenoid, anterior clinoid process, and the optic canal. It can also be used as a guide for surgery. HRCT findings indicate optic nerve injury: fractured optic canal, a hemorrhage in the ethmosphenoid sinus, optic nerve swelling, and a hematoma or emphysema in orbit. CT scans have prognostic value for patients with indirect TON [51,52].

### 4.4. MRI

MRI can be very useful in detecting nerve sheath hematomas and evaluating optic nerve integrity in trauma settings, although it is less commonly available than CT [50]. It is recommended to avoid MRI when there is a possibility of a retained metallic foreign body after trauma. Patients with indirect TON may benefit from different MRI techniques for diagnosis and follow-up. Diffusion tensor MRI (DTI) can be used to diagnose TON because

the optic nerve is a white matter fiber that arises from the axons of the RGC, and DTI is a safe and objective procedure that enables assessment of microstructure changes in white matter tract injuries [53]. Diffusion-weighted imaging of optic nerve hyperintensity can help diagnose indirect TON, according to Bodanapally et al. [54] Diffusion tensor imaging of the damaged eye did not show any results the first week after injury. Still, after the second week, it became apparent that fractional anisotropy had decreased and continued to be visible after a month. As a result of these findings, MRI is useful for detecting changes at the end of the disease process, but may not be as useful for early detection as CT [55].

### 4.5. Visual Pathway Electrophysiology

Electroretinography (ERG) measures the electrical activity of the RGCs. Two factors can cause ERG changes: changes in the refractive index in the eye and pathologic modifications in the retina. Cortical recordings of VEPs can provide information regarding dysfunctions of the visual system from the refracting media to the visual cortex. As a result, combining ERGs and VEPs makes it possible to localize functional damage between RGCs and the visual cortex [56].

### 4.6. Electroretinograms and Flash Visual Evoked Potentials

ERGs and flash-VEPs may be used to evaluate the functional status and stability of the visual pathway during craniomaxillofacial surgery and postoperative. The peak latency and amplitude of ERGs and VEPs with abnormal amplitudes (greater than 50% interocular variability) differ. The flash-VEP has three principal elements (P100, N75, and N140). There is a high degree of reliability in the P100 peak of these three components [57].

It is not necessary to use VEP to diagnose TON in all patients, but it can help determine the prognosis of patients with suspicious cases. When a patient does not remember when nerve damage occurred, when pupillary responses are unreliable, or when bilateral TON occurs, VEP has diagnostic value. VEP patients with better responses have a higher chance of recovering their vision [58,59]. Pattern reversal and flash VEP testing were used to determine the rate of vision recovery. If there is no VEP, the final vision will be poor, whereas if there is an amplitude of 50%, the vision will be good [47,59,60]. VEP has some limitations, despite its diagnostic and predictive value; for instance, it is difficult to place a VEP device on a patient with multiple trauma. Patients who suffer a concurrent brain injury may also be misdiagnosed as suffering an optic nerve injury [61].

### 4.7. Optical Coherence Tomography

The retinal nerve fiber layer has been shown to thin in patients with TON in several studies using optical coherence tomography (OCT). Due to the difficulty in locating this finding in the early stages and sitting the patient up for OCT imaging, OCT may not be useful in diagnosing TON. Optical coherence tomography is useful for long-term follow-up of optic nerve injury [62,63].

### 4.8. Doppler Sonography

Ultrasound Doppler measurements have been performed on patients with a TON to assess central retinal artery hemodynamics. There has been a decrease in systolic flow rates and end-diastolic flow rates in the patients, as well as time-averaged mean values (TAMVs) [64]. Another study confirmed this finding by showing reduced peak systolic velocity (PSV) and end-diastolic velocity (EDV) in the central retinal artery (CRA) of injured eyes [65].

### 4.9. CLOSED Protocol

Measurement of optic nerve sheath diameter (ONSD) using ultrasound can show changes to the optic nerve. A reliable and objective measurement of the nerve, sheath, and course can be obtained using arteriovenous color Doppler. A safe and accurate ON sheath evaluation can be achieved using the CLOSED protocol. The CLOSED protocol

enables operators to perform a safe and fast examination by adjusting settings to capture an image quickly and correctly [66]. Rasulo and Bertuetti recently described how color Doppler could be used to visualize retinal vessels in this regard. It is possible to reduce the likelihood of artifacts and to identify the borders of the ON sheath better when the retinal artery is visualized [67]. There is an increasing interest in the noninvasive measurement of ICP, and ONSD plays an increasingly important role in this field. Despite this, ONSD measurements are often inaccurate due to several artifacts [68].

## 5. Clinical Management of TON

A TON can be treated in four ways: conservatively, with steroids, surgically, and with a combination of surgery and steroids [41,69,70]. There is no generalized consensus on which treatment is the most favorable for TON due to the general lack of proper understanding of the exact pathophysiological mechanisms of TON.

## 6. Medical Management

### 6.1. Corticosteroids

There is experimental evidence that high-dose corticosteroids exert an antioxidant effect and thus prevent further injury after TON [71,72]. High-dose corticosteroid refers to an intravenous infusion dose of methylprednisolone at 30 mg/kg maintained by a sustained dose of 5.4 mg/kg/h for 24 or 48 h [5,73]. Microcirculation, post-injury histology, and functional outcomes were improved with methylprednisolone doses of 15–30 mg/kg in animal models of spinal cord injury [74,75]. Steroids also reduce vasospasm, edema, and nerve cell damage [76]. It was demonstrated by Sheng et al. that high-dose methylprednisolone could have an anti-apoptotic effect by upregulating Bcl-2 and downregulating Bax [77]. In a study by Anderson et al., three of the six patients treated with corticosteroids after injury recovered some vision after 6 h [76]. Seiff et al. reported no significant difference in the outcome of patients receiving steroids and others receiving surgery. Still, they noticed that the rate of improvement was much faster in those who received steroids [9]. High-dose corticosteroids seem to have more vital effects than low-dose corticosteroids in interrupting the inflammatory cascade leading to optic nerve edema [78]. It was reported that there was no significant difference in the visual outcome obtained after treatment with intravenous dexamethasone or methylprednisolone for TON [79]. Spoor et al. conducted an uncontrolled, retrospective study of 21 patients with TON. Eight patients received dexamethasone, whereas thirteen patients received methylprednisolone. About seven of nine patients in the dexamethasone group had their vision improved, while twelve of thirteen patients in the methylprednisolone group also had their vision improved [78]. The side effects of high-dose corticosteroids should always be considered.

### 6.2. Erythropoietin

Erythropoietin (EPO) is a cytokine hormone that promotes hematopoiesis, can reduce apoptosis in neurons, and provides protection in mechanical trauma, neuronal inflammation, retinal and cerebral ischemia, oxidative stress, and optic nerve injury.

Erythropoietin receptor (EPO-R) is expressed in a variety of different cells of the CNS, such as neurons, astrocytes, and oligodendrocytes. During hypoxia, neural progenitor cells release erythropoietin, stimulating neurogenesis through Epo-R [80].

Kashkouli and colleagues have shown that outcomes differ between those observed and those treated with EPO in a pilot study on traumatic optic neuropathy patients. The mean final BCVA increased significantly in the EPO group than in the observation group ($p = 0.012$). Patients with TON may benefit from intravenous erythropoietin treatment [81].

## 7. Surgical Management

Researchers found that surgery to decompress the optic nerve early can reduce axonal degeneration and improve functional outcomes in experimental rat models [82]. There needs to be more evidence that approves the maximum benefit of surgery in treating

TON [11]. There was no uniform surgical approach for treating TON until 1961 when Niho et al. designed the extracranial transethmoidal approach for decompressing the optic nerve [83]. Based on the results of the international optic nerve trauma study, there were no significant differences in the visual outcome between those treated with surgical decompression and those treated with steroids [84]. According to a recent autopsy study, there may be physical damage to the ON canal caused by trans-sphenoidal medial wall decompression with a dural sheath opening [85]. Different researchers published their surgical techniques to treat TON [86–88]. Wang Wei et al. found that endoscopic optic nerve decompression combined with corticosteroids achieved better visual outcomes in patients with no light perception [89]. In a study by Li et al., 71% of patients treated within a week of injury showed improved vision after extracranial optic nerve decompression [90].

## 8. Minimal Invasive Management

*Endoscopic Optic Nerve Decompression*

Endoscopic optic nerve decompression (EOND) is a minimally invasive procedure for optic nerve decompression in optic nerve injury patients. Trans-sphenoethmoidal and trans-sphenoid are the two main surgical endoscopic approaches. When TON does not respond to steroids, EOND may be beneficial, and if it is applied before irreversible nerve damage occurs, it may prevent permanent disability. EOND should be performed within one week after an injury in patients with confirmed optic canal fractures as a safe technique.

There is no guideline for TON treatment strategies when steroid treatment fails, and all treatments are based on specialist experience when steroid treatment is unsuccessful.

In a study, Sun et al. have shown that there were three (18.75%) cases of sixteen patients with partial recovery of VA after surgical decompression. For the remaining patients (81.25%) operating more than 10 days after injury, no improvement in VA was observed [91].

## 9. Experimental Treatments

### 9.1. Glutamate Antagonists

Glutamate is the main excitatory neurotransmitter in the nervous system, and it performs its actions by binding to NMDA receptors [92]. Memantine, phenytoin, and dizocilpine (MK801) that block the NMDA receptors may have a protective effect on the RGCs in experimental mouse models of TON [93,94]. Dexanabinol (HU-211) is a cannabinoid and a non-competitive NMDA-receptor antagonist. Dexanabinol has been shown to reduce degeneration in crush-injured rat optic nerves and promote regeneration, and this may have clinical implications [95]. Yurkewicz et al. found that traxoprodil was superior to placebo in a randomized double-blind study. Traxoprodil treatment reduced mortality by 7% compared to placebo treatment ($p = 0.08$) [96].

### 9.2. Crystallin

Crystallin is an anti-shock protein and has anti-inflammatory effects. $\alpha$-crystallin treatment suppressed TNF-$\alpha$ and iNOS expression induced by optic nerve injury in the mouse models, which are still under experimental trials [97]. Fischer et al. found that crystallin also promotes axonal regeneration [98].

### 9.3. Citrus Naringenin

Naringenin is a natural bioflavonoid found in citrus fruits [99]. It was found that both neural injury in vivo and cell stress increased JUN phosphorylation by activating the JNK-JUN pathway. JUN phosphorylation in cultured cells was completely inhibited by naringenin treatment. Chen et al., in the study using optic nerve crush injury mice, found that naringenin increased RGC survival after TON [100].

### 9.4. Anti-Inflammatory and Reactive Oxygen Species

A nitric oxide synthase (NOS) inhibitor promoted RGC survival and prevented axon degeneration [101]. A rat model of anterior ischemic optic neuropathy (rAION) that was administered granulocyte colony-stimulating factor (GCSF) found that it had antioxidant and neuroprotective properties [102]. In addition to providing an antioxidative effect, PEG-G-CSF also inhibits RGC apoptosis and neuroinflammation by activating p-Akt1/Nrf2/Sirt3 and pAkt1/Nrf2/HO-1 axes [103].

### 9.5. Nerve Growth Factors

There are several neurotrophic factors that can help restore the structural integrity of axotomized RGCs, such as fibroblast growth factor-2 (FGF-2), brain-derived neurotrophic factor (BDNF), and ciliary neurotrophic factor (CNTF) [104–106]. Chen and Weber found that BDNF could increase ganglion cell density, size, and percentage in the optic nerve-injured cat model [105] Using fibroblast growth factor in nine patients with cervical SCI was studied by Wu et al. According to the significant difference between preoperative and postoperative ASIA scores, this novel nerve repair strategy of using acidic FGF may have a role in reversing cervical spinal cord injury in humans [107].

### 9.6. Mesenchymal Stem Cells

Mesenchymal stem cells derived from human umbilical cords (hUC-MSCs) have been investigated to treat some diseases, such as spinal cord injury, Parkinson's disease, and cornea-related diseases [37,108]. A rat study has shown that intravitreal injection of human periodontal ligament-derived stem cells (PDLSCs) may improve RGC survival after optic nerve injury [109]. Twenty patients with TON were enrolled in a single-center, prospective study; group 1 received MSC local implantation with optic canal decompression; group 2 received optic canal decompression alone. There was no significant difference between groups 1 and 2 in terms of best-corrected visual acuity at follow-up ($p > 0.05$); However, group 1 had a better visual result than group 2 [110].

### 9.7. RNAs

Several pieces of evidence suggest that non-coding RNAs perform an important role in nerve regeneration such as microRNAs (miRNA), long non-coding RNAs (lncRNAs), and circular RNAs (circRNAs). The miR-19a suppresses phosphatase and tension homolog, a negative regulator of the mTOR pathway, stimulating RGC regeneration [111].

Inhibition of Semaphorin3A binding to NRP1/PlexA1 by miR-3b promotes axonal regeneration. Furthermore, miR-30b inhibited the expression of p38MAPK and activated caspase3 to reduce RGC apoptosis [112]. The upregulation of epidermal growth factor receptor pathways by inhibiting miR-21 has also been shown to promote axonal regeneration of RGCs in a rat model of optic nerve injury (ONI) [113]. Intravitreal injection of siRNA against caspase-2 has not been found to protect RGCs from ONI in the mouse model [114]. However, in ONI therapy, miRNAs remain a promising tool.

### 9.8. Hypothermia

Hypothermia is frequently applied in medicine to prevent damage to organs or tissues. Hypothermia helps protect RGCs, especially in the context of ischemia, according to large numbers of animal studies [115]. The use of hypothermia for the treatment of traumatic vision impairments can be very helpful. Researchers found that mild hypothermia treatment (32–34°C) significantly impacted neuroelectrophysiological outcomes compared with normothermia treatment in 44 patients with brain injury [116].

### 9.9. Lipids

Lipids compose the majority of neuronal structures and functions. As a result of neuronal injury, lipids undergo various changes [117,118]. Coordination of changes in lipid homeostasis is required for axonal regeneration in damaged neurons [119].

Research on the regulation of lipid metabolism in neurons undergoing regeneration is limited [120].

Researchers found that resveratrol treatment improved levels of cholesterol in RGCs and increased the neuroprotective properties of damaged RGCs in TON [121].

Further studies have shown that extracellular signal-regulated kinase 1/2 mediates neuroprotective and regenerative effects in impaired RGCs and neurons [122,123]. Axonal regeneration and injury are both mediated by lipid signal transduction pathways [124].

Stark D.T. et al. demonstrated that the matrix-aided laser ionization technique improved nerve regeneration in ONI by increasing lysosome-specific lipids and GM3 gangliosides. Findings provided new insights into the lipid changes that occur after ONI and the regeneration of axons. However, more research is needed on lipids and their role in TON [125].

### 9.10. Mitotherapy

Mitotherapy (transplantation of exogenous mitochondria) has recently been shown to mitigate neurological disease progression [126,127]. Researchers found that after intravitreal transplantation, integrating active mitochondria enabled the retina to improve its oxidative metabolism and electrical activity one day after transplantation. Additionally, mitotherapy led to an increased axonal extension of the injured neurons at 28 days, as well as an increase in RGC survival at 14 days [128]. This study supports mitotherapy as a potential therapeutic intervention for TON.

## 10. Conclusions

TON often results in permanent vision impairment, affecting the patient's quality of life for the rest of their lives. Providing prompt and accurate diagnoses to the patient can save their vision. Patients can usually be diagnosed based on their clinical symptoms (i.e., loss of vision, VF defects, impaired color vision, and relative afferent pupillary defect). CT scans can be used for small fractures of clinical doubt. DTI can assist in diagnosing TON (with HRCT or MRI) and can contribute information about the pathophysiological process of TON. Whenever clinical evaluation is unreliable (consciousness and distinct periorbital swelling), VEP and ERG may assist surgeons in deciding whether surgery is necessary. The prognostic value of flash-evoked ERGs for visual acuity recovery is high. Therefore, in the absence of an ERG, visual acuity does not return. The most effective treatments for TON have been steroids and decompression surgery. Multiple trauma patients cannot undergo multiple surgeries. Several additional treatments are being utilized to preserve and restore the optic nerve. To establish a standard treatment protocol, it is necessary to conduct large randomized controlled trials. Research on new drugs, mechanisms, and techniques (such as glutamate antagonists, crystallin, citrus naringenin, targeting anti-inflammatory and reactive oxygen species, nerve growth factors, mesenchymal stem cells, RNAs, hypothermia, lipids, and mitotherapy) also provides new ideas for treating TON.

**Author Contributions:** B.L.-W. was the major contributor to the design of the study. M.R.H.S. and MA were responsible for writing the first manuscript. M.R.H.S. was responsible for revising the manuscript and validating the included studies. M.R.H.S. and M.A.A. contributed editing graphs. B.L.-W. conceived the study and was in charge of overall direction and planning. All authors contributed to the article and approved the submitted version. All authors have read and agreed to the published version of the manuscript.

**Funding:** This research received no external funding.

**Institutional Review Board Statement:** Not applicable.

**Informed Consent Statement:** Not applicable.

**Data Availability Statement:** Not applicable.

**Conflicts of Interest:** The authors declare no conflict of interest.

## Abbreviations

Traumatic optic neuropathy (TON); Retinal ganglion cell (RGC); Reactive oxygen species (ROS); Nerve fiber layer (NFL); Nitric oxide synthase (NOS); Fibroblast growth factor-2 (FGF-2); Brain-derived neurotrophic factor (BDNF); Ciliary neurotrophic factor (CNTF); Optic nerve injury (ONI); Erythropoietin (EPO); Erythropoietin receptor (EPO-R); Endoscopic optic nerve decompression (EOND); Relative afferent pupillary defect (RAPD); Pattern-reversal visual evoked potentials (P-VEPs); Electroretinography (ERG); Optical coherence tomography (OCT); Time-averaged mean values (TAMVs); Central retinal artery (CRA); Peak-systolic velocity (PSV); End-diastolic velocity (EDV); Optic nerve sheath diameter (ONSD); Visual acuity (VA); Visual field (VF); Aquaporin 4 (AQP4).

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
