# Peer review of "Traumatic Optic Neuropathy: Update on Management"

_encyclopedia, doi:10.3390/encyclopedia3010007_

Round 1

Reviewer 1 Report

Q1 Traumatic optic neuropathy (TON) is a grave complication of closed head injuries. It  accounts for about 0.5–5% of all closed-head injuries.1(superscript ï¼‰

Q2 The visual loss may manifest sev- 19 eral months after trauma; therefore, the diagnosis is always delayed.2(superscript ï¼‰

Q3 In Figure 1, the authors included inflammatory mediators, which did not discuss in the context.

Q4 Figure 2. Nothing addressed in the context regarding Figure 2.

Q5 Line 231: 17. Targeting Anti-inflammatory and ROS;

Q6 conclusion: relatively weak, suggest to rewrite.

Reviewer 2 Report

Really well written paper about this topic which is really fascinating. I've some advices for you:

- explain better optic ultrasound evaluation which is really to predict secondary optic injuries due to intracranial hypertension (also using CLOSED protocol) (1) (2)

- explain dosage of steroids used in optic trauma management (e.g. 10 mg/kg for desametasone etc). (3)

Anyway I want to thank your efforts writing this topic which is not so known and really usefull. 

Bibliography: 

1. Aspide R, Bertolini G, Albini Riccioli L, Mazzatenta D, Palandri G, Biasucci DG. A Proposal for a New Protocol for Sonographic Assessment of the Optic Nerve Sheath Diameter: The CLOSED Protocol. Neurocrit Care. 2020 Feb;32(1):327-332. doi: 10.1007/s12028-019-00853-x. PMID: 31583527.

2. https://www.ncbi.nlm.nih.gov/books/NBK554479/

3. https://www.ncbi.nlm.nih.gov/pmc/articles/PMC7390489/

Best regards

Author Response

Point 1: - explain better optic ultrasound evaluation which is really to predict secondary optic injuries due to intracranial hypertension (also using CLOSED protocol) (1) (2)

Response 1: I added the section as Diagnosis:

4. Diagnosis

4.1. Clinical manifestation

Most acute or delayed visual function disorders associated with TON cannot be explained by anything other than a blunt craniofacial injury.

The following clinical features characterize TON:

(1) An immediate or delayed loss of vision after an injury,

(2) VF defects,

(3) Impaired color vision,

(4) relative afferent pupillary defect (RAPD): occurs when the direct light reflex delays or disappears compared to the indirect light reflex; known as Marcus-Gunn pupil.5,38 Afferent disorders of the visual pathway are often associated with frontonasoethmoidal fractures.42

Traumatic optic neuropathy is often associated with frontal (72%) or frontotemporal (12%) craniocerebral injuries. The most common causes are traffic accidents and falls.43

An abnormal direct light reflex provides the most reliable diagnosis in patients with impaired consciousness. Most patients undergo a fundus examination, initially appearing normal if the lesion is not involving the retina and/or anterior optic nerve. However, the patient's optic disc will later become pale and atrophied.44

The diagnosis of TON may be delayed by several combined craniofacial or other systemic injuries. The diagnosis may also be complicated by non-ON damage (damage to the eye or intracerebral visual circuits, etc.). In this case, the diagnosis of TON requires a comprehensive clinical assessment (ranging from ophthalmic to radiographic assessments).45

The majority of patients with TON present to the clinic with vision loss. The patient should undergo a complete ophthalmologic evaluation, including VF, VA, and fundus photography.46 If the clinical ophthalmologic evaluation of the eye is not possible, pattern-reversal visual evoked potentials (P-VEPs) can be used. The P-VEP is useful in assessing visual function in children and infants as well as detecting nonorganic vision loss, nonorganic vision loss in complex patients, and optic neuropathy. 47

4.2. Imaging

The aim of imaging is to determine the extent of damage to the visual pathway. Meanwhile, traditional radiographs have less valuable. The imaging modalities of computed tomography (CT) and magnetic resonance imaging (MRI) are preferred by some physicians when treating patients who are progressively visually impaired or who require therapeutic intervention. 48,49 Radiological investigations should be performed immediately for patients with head or oculofacial trauma and optic nerve damage (one or both decreased VAs, VF defects, and afferent pupillary defects).50

4.3. CT Scan

HRCT is a standard imaging technique for evaluating suspicious patients of TON because that can display a fracture of the ethmoid, sphenoid, anterior clinoid process, and the optic canal. It can also be used as a guide for surgery. HRCT findings indicate optic nerve injury: fractured optic canal, a hemorrhage in the ethmosphenoid sinus, optic nerve swelling, and a hematoma or emphysema in orbit. CT scans show prognostic value for patients with indirect TON and posterior orbital fractures, compared to indirect TON and anterior orbital fractures.51,52

4.4.MRI

MRI can be very useful in detecting nerve sheath hematomas and evaluating optic nerve integrity in trauma settings, although it is less commonly available than CT.50 It is recommended to avoid MRI when there is a possibility of a retained metallic foreign body after trauma. Patients with indirect TON may benefit from different MRI techniques for diagnosis and follow-up. Diffusion tensor MRI (DTI) can be used to diagnose TON because the optic nerve is a white matter fiber that arises from the axons of the RGC, and DTI is a safe and objective procedure that enables assessment of microstructure changes in white matter tract injuries.53 Diffusion-weighted imaging of optic nerve hyperintensity can help diagnose indirect TON, according to Bodanapally et al.54 Diffusion-tensor imaging of the damaged eye did not show any results the first week after injury. Still, after the second week, it became apparent that fractional anisotropy had decreased and continued to be visible after a month. As a result of these findings, MRI is useful for detecting changes at the end of the disease process, but may not be as useful for early detection as CT.55

4.5. Visual Pathway Electrophysiology

Electroretinography (ERG) measures the electrical activity of the RGCs; Two factors can cause ERG changes: changing the refractive index in the eye and pathologic modifications in the retina. Cortical recordings of VEPs can provide information regarding dysfunctions of the visual system from the refracting media to the visual cortex. As a result, combining ERGs and VEPs makes it possible to localize functional damage between RGCs and the visual cortex.56

4.6. Electroretinograms and Flash Visual Evoked Potentials

ERGs and flash-VEPs may be used to evaluate the functional status and stability of the visual pathway during craniomaxillofacial surgery and postoperative. The peak latency and amplitude of ERGs and VEPs with abnormal amplitudes (greater than 50% interocular variability) differ. The flash-VEP has three principal elements (P100, N75, and N140). There is a high degree of reliability in the P100 peak of these three components.57

It is not necessary to use VEP to diagnose TON in all patients, but it can help determine the prognosis of patients with suspicious cases. When a patient does not remember when nerve damage occurred, when pupillary responses are unreliable, or when bilateral TON occurs, VEP has diagnostic value. VEP patients with better responses have a higher chance of recovering their vision.58,59 Pattern reversal and flash VEP testing were used to determine the rate of vision recovery. If there is no VEP, the final vision will be poor, whereas if there is an amplitude of 50%, the vision will be good.47,59,60 VEP has some limitations, despite its diagnostic and predictive value. For instance, multi-injury patients in special wards are difficult to place a VEP device. Patients who suffer a concurrent brain injury may also be misdiagnosed as suffering an optic nerve injury.61

4.7. Optical Coherence Tomography

 The retinal nerve fiber layer has been shown to thin in patients with TON in several studies using optical coherence tomography (OCT). Due to the difficulty in locating this finding in the early stages and sitting the patient up for OCT imaging, OCT may not be useful in diagnosing TON. Optical coherence tomography is useful for long-term follow-up of optic nerve injury.62,63

4.8. Doppler Sonography

Ultrasound Doppler measurements have been performed on patients with a TON to assess central retinal artery hemodynamics. There has been a decrease in systolic flow rates and end-diastolic flow rates in the patients, as well as time-averaged mean values (TAMVs).64 Another study confirmed this finding by showing reduced Peak-systolic velocity (PSV) and end-diastolic velocity (EDV) in the central retinal artery (CRA) of injured eyes.65

4.9. CLOSED protocol

Measurement of optic nerve sheath diameter (ONSD) using ultrasound can show to change of optic nerve. A reliable and objective measurement of the nerve, sheath, and course can be obtained using arteriovenous color Doppler. A safe and accurate ON sheath evaluation can be achieved using the CLOSED protocol. The CLOSED protocol enables operators to perform a safe and fast examination by adjusting settings to capture an image quickly and correctly.66 Rasulo and Bertuetti recently described how color Doppler could be used to visualize retinal vessels in this regard. It is possible to reduce the likelihood of artifacts and to identify the borders of the ON sheath better when the retinal artery is visualized.67 There is an increasing interest in the noninvasive measurement of ICP, and ONSD plays an increasingly important role in this field. Despite this, ONSD measurements are often inaccurate due to several artifacts.68

Point 2: explain dosage of steroids used in optic trauma management (e.g. 10 mg/kg for desametasone etc). (3)

Response 2: High-dose corticosteroid refers to an intravenous infusion dose of methylprednisolone at 30 mg/kg maintained by a sustained dose of 5.4 mg/kg/h for 24 or 48 hours.

Reviewer 3 Report

In this manuscript the authors discussed pathophysiologic mechanisms of traumatic optic neuropathy, available and future treatments. Some concerns are listed as below:

Please provide the full name of TON in the abstract.

The subtitles of this review need modification. 

The description in some sections is too simple.

When it comes to corticosteroids, it is not clear how long corticosteroids should be treated.

The biggest issue is that the review is simply a repetition of the literature with no attempt to synthesize or critically discuss the results presented. The description in some sections is too simple in this paper. For example, it is not clear for readers when or how long corticosteroids should be treated in the condition of traumatic optic neuropathy. Regarding other emerging treatments, the evidence in the manuscript can not support the conclusions. Potential side effects should not be ingored in this paper.

Round 2

Reviewer 1 Report

I am satisfied with the authors' hard work and good revision according to my suggestions. I recommend it to be  accepted.

Author Response

Thank you.

Reviewer 2 Report

Perfect

Author Response

Thank you.

Reviewer 3 Report

The authors have answered my questions. I would suggest to invite another reviewer to go through this manuscript.

Author Response

Thank you.